# A Personalized Longitudinal Strategy in Low-Grade Glioma Patients: Predicting Oncological and Neural Interindividual Variability and Its Changes over Years to Think One Step Ahead

**DOI:** 10.3390/jpm12101621

**Published:** 2022-10-01

**Authors:** Hugues Duffau

**Affiliations:** 1Department of Neurosurgery, Gui de Chauliac Hospital, Montpellier University Medical Center, 80 Av. Augustin Fliche, 34295 Montpellier, France; h-duffau@chu-montpellier.fr; Tel.: +33-4-67-33-66-12; Fax: +33-4-67-33-69-12; 2Team “Plasticity of Central Nervous System, Stem Cells and Glial Tumors”, National Institute for Health and Medical Research (INSERM), U1191 Laboratory, Institute of Functional Genomics, University of Montpellier, 34091 Montpellier, France

**Keywords:** low-grade glioma, awake surgery, brain mapping, quality of life, overall survival, interindividual variability, connectome, neurocognition, personalized management, long-term outcomes

## Abstract

Diffuse low-grade glioma (LGG) is a rare cerebral cancer, mostly involving young adults with an active life at diagnosis. If left untreated, LGG widely invades the brain and becomes malignant, generating neurological worsening and ultimately death. Early and repeat treatments for this incurable tumor, including maximal connectome-based surgical resection(s) in awake patients, enable postponement of malignant transformation while preserving quality of life owing to constant neural network reconfiguration. Due to considerable interindividual variability in terms of LGG course and consecutive cerebral reorganization, a multistage longitudinal strategy should be tailored accordingly in each patient. It is crucial to predict how the glioma will progress (changes in growth rate and pattern of migration, genetic mutation, etc.) and how the brain will adapt (changes in patterns of spatiotemporal redistribution, possible functional consequences such as epilepsy or cognitive decline, etc.). The goal is to anticipate therapeutic management, remaining one step ahead in order to select the optimal (re-)treatment(s) (some of them possibly kept in reserve), at the appropriate time(s) in the evolution of this chronic disease, before malignization and clinical worsening. Here, predictive tumoral and non-tumoral factors, and their ever-changing interactions, are reviewed to guide individual decisions in advance based on patient-specific markers, for the treatment of LGG.

## 1. Introduction

Diffuse low-grade glioma (LGG), i.e., WHO grade II glioma [1], is a rare brain cancer, whose ethiopathogeny is poorly understood, making difficult the prediction of its natural course, especially at an individual level [2]. LGG spontaneously exhibits different stages in its evolution, namely (i) a pre-symptomatic period in which the tumor is usually slow-growing, as demonstrated in cases of incidental discovery [3]; (ii) a symptomatic period in which the glioma induces clinical consequences, usually seizures and/or mild cognitive impairments visible on neuropsychological assessment [4], while continuing to progress slowly but constantly (about 3–4 mm mean diameter per year) [5]; and (iii) a period of malignant transformation (MT) with acceleration of the growth rate, resulting in more severe neurological deficits and ultimately death [6]. However, LGG represents a heterogeneous group of tumors with various courses, which are difficult to predict at the individual level and at each stage of the disease. In addition, due to its unknown origin and diffuse features within the central nervous system, this is an incurable tumoral disease. Thus, the main goal of therapeutic management is to delay progression to a higher grade of malignancy while preserving quality of life (QoL) for as long as possible [7]. To this end, early maximal safe surgical excision represents initial treatment, since reduction of the tumor volume decreases the risk of MT [8] and thus prolongs overall survival (OS) [9,10,11,12,13]. Moreover, because intratumoral heterogeneity is frequent in LGG, large resection increases the chances of detecting possible microfoci of high-grade glioma within the neoplasm, enabling better adaptation of the next strategy according to extensive histomolecular data [14,15,16]. In other words, from an oncological perspective, variability in LGG is considerable not only at any given moment, then over months or years, but also from one patient to another one depending on the intrinsic glioma behavior (partly related to the genetic subtype), and depends also on the time of diagnosis in the natural history of the disease [17].

In the same spirit, with regard to functional considerations, there are very high variations between brain organization across patients as well as in the same patient over time, due to physiological interindividual anatomo-functional variability, which is increased in the event of LGG [18]. In fact, this type of slow-growing neoplasm which progresses over years or even decades [19] may induce reactional processes of neural network reconfiguration that allow functional compensation, at least at the first stage of the disease [20]. This explains why patients harboring LGG, mostly young adults, usually enjoy active familial and socio-professional lives, sometimes despite the presence of a voluminous tumor at diagnosis, possibly involving so-called “eloquent” areas. In practice, it is essential to better explore the mechanisms underpinning neuroplasticity at the individual level, since the patterns of cerebral reallocation may vary from one patient to another (for example, predominant compensatory recruitment of peritumoral structures versus predominant recruitment of contra-hemispheric homologous areas [21]), as well as in a given patient over time—e.g., with a switch from a perilesional functional rearrangement to a more bilateral reshaping [22]. Prediction of such potential for metaplasticity (plasticity of plasticity) is very complex, while the therapeutic strategy should be tailored according to these brain connectome dynamics for each individual glioma patient [23]. This is particularly relevant for planning maximal surgical resection, which must be achieved up to the functional boundaries identified by intraoperative mapping in conscious patients, with the aim of preserving cortico-subcortical networks critical for neural functions at the individual level [24]. Considering that the possibility of reoperation several months or years later will depend on the way the brain has reorganized (or not) since previous surgery, if optimization of the resection does not seem feasible because the limitations of neuroplasticity have been reached, medical adjuvant treatment represents a relevant alternative [25].

Consequently, in order to elaborate a multistage longitudinal personalized approach, it is of utmost importance to predict how the glioma will progress, in terms of possible changes in its growth rate, pattern of migration, genetic mutation, etc., and how the brain will adapt through changes in its pattern of spatiotemporal redistribution or possible functional consequences such as epilepsy or cognitive decline, etc. Indeed, because diffuse LGG inevitably recurs, and because the therapeutic armamentarium is not inexhaustible, the main goal in this respect if to anticipate the requirements of individualized management. In this way, practitioners can remain one step ahead in order to select the optimal surgical and/or medical oncological (re-)treatment(s)—some of them possibly kept in reserve for years—at the appropriate time in the evolution of this chronic disease, before MT and before clinical worsening. The purpose of this current study is to review tumoral and non-tumoral predictive factors, as well as their interactions, which might be helpful to guide individual decisions months or years in advance, to establish personalized multistep strategies for LGG patients who currently have a life expectancy of over 15 years [12,26].

## 2. Predicting Oncological Interindividual Variability and Its Changes over Time

Distinct types of factors must be taken into consideration, including those related to the tumoral disease itself as well as external factors which may influence the course of glioma (Figure 1).

### 2.1. Factors Related to the Glioma

#### 2.1.1. Tumoral Volume

The volume of LGG at diagnosis is highly variable, from 0.39 to 386 cc according to recent studies [4,27]. Notably, it has long been considered that a larger size of LGG is an adverse prognostic factor; however, only one dimension of the tumor was classically measured, with a poorer prognosis when the size was ≥6 cm [28,29,30]. More recently, it has been evidenced that a greater LGG volume (by definition, calculated in 3D) was significantly correlated with a higher risk of MT and with shorter OS [9,10,26]. This may be explained by the fact that a more voluminous glioma reflects a more prolonged natural history of the disease, with a higher number of tumoral cells and therefore an increased risk of mutational accumulations [31,32]. Such a hypothesis is supported by a decreased risk of MT as well as a significant survival benefit in incidental LGG compared with symptomatic LGG discovered later [3], although a higher volume at discovery of incidental LGG is also significantly associated with more progressive tumors [33]. Consequently, a screening policy has been proposed to diagnose and treat LGG earlier, as well as to optimize the opportunity to better understand the origin of these tumors [34]. In addition, during the follow-up of LGG patients already treated, it has also been suggested that further treatment(s) should be considered when the tumor volume re-increases over time, especially when it reaches a threshold of about 10–15 cc (even if the patient is asymptomatic at that time), in order to delay MT [7].

#### 2.1.2. Growth Rate

The velocity of a tumor’s spontaneous expansion, which can be plotted as a function of mean glioma diameter over time (computed from the volumes calculated by repeat MRIs) is predictive of long-term outcomes for LGG [5]. Indeed, the slope of the mean tumor diameter growth curve is an independent prognostic factor for malignant progression-free survival and for OS as a continuous predictor—that is, showing a linear relationship between OS and growth rate [35]. The relevant kinetics are very variable from one LGG to another at diagnosis, from less than 1 mm per year to 8 mm per year; over 8 mm/year, the LGG invokes a greater risk of MT [35]. Interestingly, this marker is independent of the molecular profile [35], in particular with regard to the IDH status [36]. Thus, identifying rapidly growing LGG during the pretherapeutic period, i.e., tumors at higher risk of worsening evolution, can be helpful to decide when to start treatment(s) [37], including for incidental LGG [38]. This is also true during the surveillance of LGG patients already treated, in order to determine when to consider further therapy. Especially because the growth rate is similar before and after surgery [39], acceleration of the velocity of the residual tumor in cases of incomplete resection may support earlier re-operation, even in asymptomatic patients [25]. Furthermore, re-growth after a period of stabilization following chemotherapy and/or radiotherapy can prompt consideration of further treatment [7]. The possibility of such changes favors deployment of a systematic control MRI every 3 to 6 months throughout life.

#### 2.1.3. Pattern of Migration Versus Proliferation and Tumor Location

In addition to variable glioma volumes and kinetics, LGG may exhibit distinct patterns of progression within the brain, i.e., a more proliferative “bulky” pattern versus a more diffuse “migratory” one. Interindividual variability may be considerable, ranging from a very focal tumor to a gliomatosis with bi-hemispheric dissemination [40]. Notably, in incidentally discovered focal LGG, the insular location was found to be a predictive factor of a more progressive tumor [33]. In a study of 1097 cases, a tumor located in a nonfrontal area was an independent factor of poor prognosis [10]. With its invasive profile, glioma has a high propension to migrate along the subcortical fibers, as shown radiologically [41,42] as well as pathologically: biopsy samples evidenced that the tumor cells followed the white matter tracts and were slightly more concentrated in the peripheral parts of those tracts [43]. It seems that the myelin status could play a pivotal role not only in LGG invasion in adults but also possibly in its origin in teenagers [44]. Biomathematical models have attempted to anticipate the profile of glioma evolution [45], knowing nonetheless that this pattern might change over time, maybe at least partly in relation to the therapeutic effects; for example, a bulky LGG before surgery may switch towards a predominantly diffuse pattern after incomplete resection [40]. Such a parameter is critical for treatment selection, because white matter connectivity represents the main limitation of neuroplasticity (see below). This could be a major problem for achieving massive surgical resection or for wide brain radiation therapy if the patient hopes to preserve an optimal QoL, particular in very diffuse LGG [23].

#### 2.1.4. The Peritumoral Zone

LGG is a heterogeneous and poorly circumscribed neoplasm with isolated tumor cells (ITC) that extend beyond the margins of the lesion depicted on MRI, as demonstrated by biopsy samples taken from within and beyond the “glioma core” (visible as a T2-FLAIR hypersignal on MRI); ITCs have been detected behind such signal abnormalities [43,46]. It is worth noting that the cycling tumor cell fraction was higher at the limits of the MRI-defined abnormalities than when closer to the center of the tumor, in 62.5% of patients [47]. This could explain the high risk of glioma relapse at the periphery of the surgical cavity, even following large resection [48]. Efforts to demarcate the glioma core from the surrounding healthy brain led to the definition of an intermediate region, the so-called peritumoral zone (PTZ) [49]. An important interindividual variability exists regarding this PTZ, as demonstrated by samples which found ITC from 10 to 20 mm around the glioma core [43,46], in agreement with the fact that the tumor core might be more bulky or more diffuse (see above). Interestingly, recent investigations have indicated that this interface between the glioma core and the healthy brain represents a specific metabolic and cellular entity. Such characteristics of the PTZ that are being increasingly explored through radiomics and radiogenomics [50,51] may play a pivotal role for decision making in the management of diffuse LGG.

#### 2.1.5. Metabolic Changes

While still a matter of debate from a radiological perspective, the occurrence of an enhancement during the course of LGG is usually associated with MT [52,53]. However, if the tumor has already become more aggressive when the (re)treatment is proposed, this means in essence that the opportunity for action has been missed [38]. Therefore, since the main oncological goal is to prevent MT, additional non-invasive metabolic information may be useful in order to predict when the LGG has a higher risk of degeneration. First, an increase of perfusion or diffusion value(s) obtained through sequential and multimodal MRI could be a predictor of changes in glioma behavior [54], possibly identifiable using new machine-learning classifiers [55], and might prompt earlier (re)treatment. In the same spirit, recent advances in PET scanning using tracers easily accessible in routine practice (such as F-DOPA) have enabled an increase in sensitivity for the detection of foci of MT within the LGG, before the onset of enhancement [56]. As mentioned, metabolic imaging could also be helpful to better investigate the PTZ [49]. This additional information could be of utmost interest for deciding the best timings of new therapies during follow-up of LGG patients.

#### 2.1.6. Multicentric LGG and Leptomeningeal Dissemination

Although LGG is usually a solitary tumor (with a unique location in the brain), multicentric glioma may exist [57]: this is a scarce occurrence in which the tumors are in different lobes or hemispheres and are completely separated, with no anatomical continuity between them [58]. These gliomas can be synchronous or can exhibit metachronous development at different times during the course of the disease [59]. Indeed, after surgery for a solitary LGG, the emergence of remote gliomas may be observed in the same hemisphere as the initial LGG, in the contralateral hemisphere, or even in the posterior fossa, regardless of the possible local relapse of the initial tumor or otherwise; this second tumor can be an LGG or a high-grade glioma [40]. Although the pathophysiology of this rare progression remains poorly understood, therapeutic management should be adapted accordingly in the knowledge that multicentric LGG can be removed safely, supporting surgery as the first treatment, as in solitary LGG [57,59]. In addition, adjuvant therapies can be considered, especially in the event of metachronous high-grade glioma emerging away from the initially resected LGG [40]. In practice, this means that an accurate examination of the whole brain must be achieved at each MRI control.

In a similar context, another pattern of non-locoregional progression of LGG is leptomeningeal dissemination (LMD), previously rarely described [60] but currently more often encountered due to improvements in the focal control of the glioma [40]. In fact, as for the onset of metachronous multicentric LGG, LMD may occur even in cases of non-progression of the initial tumor [60]. In other words, it seems that the long-term course of LGG can be modified to obtain optimization of the therapeutic strategy (at least locally), resulting in a more global dissemination of the disease; this should encourage neurooncologists to think on a larger scale.

#### 2.1.7. Histomolecular Profile and Intratumoral Heterogeneity

Although LGG is traditionally classified according to its cell morphology (astrocytes, oligodendrocytes, or a combination of both), considerable technical advances in the field of molecular biology, such as DNA methylome profiling, have permitted the introduction of new tumor types and subtypes. Recent advances allow identification of genetic factors valuable for predicting LGG behavior—in particular 1p19q status (a marker of better prognosis when co-deleted), IDH status (a marker of better prognosis when mutated), MGMT status (a marker of better prognosis when methylated), TERT status, etc. [61]. These markers led to the refinement of the WHO classification based on an “integrated diagnosis” [1], critical for therapeutic decision making according to the current guidelines [62]. However, although statistically relevant at a group level, each particular factor is unreliable at the individual level; for example, an IDH-mutated LGG can nonetheless have a rapid growth-rate with poor intrinsic prognosis [36], whereas patients with IDH wild-type gliomas may have long survival, especially after large surgical resection [63].

Moreover, the WHO classification does not take into account the existence of a major intratumoral heterogeneity which is very frequent in LGG. As demonstrated by multiple samples or when extensive surgical resections have been performed “en bloc”, distinct histo-molecular components have often been observed within the same tumor [14,15,16,31,64,65]. Moreover, recent research using intraoperative image-guided biopsies, genetic analyses with RNA sequencing, and whole-exome sequencing reported a gene expression pattern and mutational landscape of the PTZ that were distinct from those seen in the tumor core and peripheral brain tissue [66]. In addition, in cases of reoperation(s), updated neuropathological examinations may show changes in the grade of malignancy as well as in the molecular profile [32]. As a consequence, LGG genetics represents only part of the history, and caution should be exercised before applying a therapeutic strategy on the basis of molecular markers, which risks overlooking other critical clinical and radiological factors [67,68].

### 2.2. External Factors to the Glioma

#### 2.2.1. Familial Predisposition

While the majority of gliomas are sporadic in origin, familial gliomas have been described, although these are exceptionally rare, especially in the form of LGG [69,70]. A potential heritable etiology for glioma families has been evoked; specifically, high-penetrance familial mutations and common low-penetrance susceptibility loci (e.g., single-nucleotide polymorphisms (SNPs)) may contribute to familial glioma risk [70,71]. Nonetheless, recent series have shown that familial gliomas, including LGG, showed similar genomic and molecular biomarker profiles to sporadic gliomas, consistent with the similarity in their clinical features [72,73]. However, identification of new susceptibility factors in familial LGG might help to elucidate the molecular pathogenesis of gliomas [73]. In practice, to increase the chances of earlier diagnosis of possible LGG, screening can be offered to relatives of gliomas patients, and such an “intentional discovery” may lead to more rapid treatment in the first period of the disease [74].

#### 2.2.2. Age

Older age has been correlated with poorer oncological outcomes in LGG patients, even though the cut-off may vary, e.g., 40 years [28,29,30] versus 55 years [10]. Nevertheless, even if the prognosis seems to be directly linked to age per se, it cannot be rule out that glioma discovery in a younger patient may mean that the diagnosis was made at an earlier stage of the tumoral disease—therefore, with lesser volume and fewer mutational changes. Furthermore, LGG mainly affects young adults, explaining why screening policy design has suggested the application of MRI in a selected population before 40 years [75]. In clinical routine, beyond the age at diagnosis and during the years (or even decades) of LGG management, practitioners be aware that the risk of MT is potentially increasing as the patient becomes older, justifying continuation of regular surveillance even in cases of long-term tumor stabilization [40].

#### 2.2.3. Pregnancy

Another factor which could change the natural course of LGG is pregnancy. Indeed, previous studies have demonstrated that pregnancy might facilitate LGG progression, with an increase of growth rate and a higher risk of MT, as well as earlier clinical deterioration [76,77,78]. In a recent cohort of women who were pregnant after LGG resection and with stable oncologic disease at the time of pregnancy, 43.7% of patients had an LGG which changed its behavior (i.e., with an acceleration of the diameter expansion velocity and/or the onset of a contrast enhancement) during or within the 3 months following pregnancy. The median time of death was 3.9 years from delivery, with an OS of only 5.7 years from delivery across the whole study [79]. Remarkably, postoperative tumor residual volume and tumor velocity before pregnancy were significant predictive markers conditioning post-pregnancy survival, indicating strong interactions between intrinsic and external glioma factors. Therefore, identifying patients at risk is crucial for providing relevant counselling to women experiencing LGG who have a desire for motherhood, and it is important to act accordingly, for example, to propose reoperation on a patient with a postsurgical residue in order to complete radical resection before the beginning of pregnancy [25].

#### 2.2.4. Therapeutic Factors and LGG Behavior

Surgery: early maximal resection is the first treatment to propose at diagnosis in LGG patient, as it is correlated to an increased OS [9,10,11,12,13]. This is particularly true following gross total resection, which was significantly associated with decreased mortality and likelihood of progression at all time points compared with subtotal resection, according to a recent meta-analysis [80]. One step forward is to achieve a “supra-complete” resection, to remove the PTZ which contains ITC exposing patients to a high risk of recurrence [49,66]. Long-term follow-up after supra-marginal LGG excision has shown a stronger impact on the natural course of the disease [27,48,81,82]. As a consequence, it has been proposed to consider LGG as remaining “high-risk” even in young patients (<40 years) who have had only a subtotal resection [83]. It seems that this oncological benefit from surgery is related to a cytoreductive effect, i.e., the decrease of the tumor volume reduces the risk of MT, as supported by recent literature in which longer malignant progression-free survival was reported after more extensive resection. This is in agreement with the prognostic role of the preoperative tumor volume, previously discussed (for reviews on this topic, see [84,85]). Interestingly, recent series have demonstrated that maximal surgical resection was strongly associated with more favorable outcomes regardless of molecular subtypes [11,27,63,86]. Moreover, in cases of foci of high-grade gliomas (III/IV) in the middle of the tumor, if maximal surgical excision has been achieved, the postoperative course of the neoplasm can continue to correspond with LGG behavior. Therefore, postponing adjuvant treatment can be envisioned, at least in glioma with a slow growth rate (regardless of its molecular profile), which was found to result in >95% survival at 5 years [87].

Importantly, although the extent of resection (EOR) (percentage of glioma removed) was for a long time considered as the main predictive factor, the calculation of the postoperative tumoral volume (if any) indicated on the FLAIR-weighted MRI is a valuable predictive factor of LGG re-progression [10,88,89]. Therefore, glioma residual volume should be objectively calculated in a systematic way following each LGG surgery.

From the same point of view, early reoperation(s) at the time of glioma relapse should be systematically considered before MT. Indeed, a recent review evidenced that multiple surgeries were associated with prolonged OS while preserving quality of life [25]. In particular, when three repeat LGG resections were performed, Hamdan et al. reported an unprecedent OS of almost 18 years [12]. Taken as a whole, these results support the idea of radical and “prophylactic” surgery, which led to the proposal of early resection in incidental LGG. This new approach achieved more than 60% supra-complete or total resections with a survival rate of over 93% after 10 years [13], while preserving quality of life [90,91]. These recent findings support the screening policy previously discussed [34,75].

Adjuvant medical treatments: Chemotherapy alone can be considered as initial treatment for LGG, especially in oligodendroglioma [92]. Indeed, a recent high-quality quantitative review evidenced that chemotherapy in LGG was associated with decreased mortality at 5 and 10 years [80]. Conversely, early radiation therapy (RT) was not associated with decreased mortality—even though progression-free survival (PFS) was improved compared with patients receiving delayed or no radiation [29,80], in the knowledge nonetheless that PFS is only an imperfect surrogate of long-term survival [93]. Combination of RT with procarbazine, CCNU (lomustine), and vincristine (PCV) resulted in prolonged OS in comparison with RT alone [83]. However, due to its possible negative impact on long-term cognition [94], this regimen has not imposed itself as the gold-standard treatment after surgery, especially in the European Low-Grade Glioma Network, in which Temozolomide is largely used as first-line treatment after surgical resection for high-risk LGG patients, or at progression [95]. Interestingly, predictive factors of chemosensitivity have been identified, especially 1p19q co-deletion and MGMT methylation, to select patients who can benefit from upfront treatment with Temozolomide, with the aim to defer radiation therapy to enable long-lasting preservation of quality of life [96]. In case of non-(re)operable LGG, it was even proposed to administrate neoadjuvant chemotherapy to induce a shrinkage of the glioma and then to (re)open the door for subsequent surgery with an optimized EOR [97,98]. Importantly, because responses to Temozolomide or PCV remain highly variable across patients, and in the same patient when receiving a second line of chemotherapy, in addition to the use of multimodal imaging [99], data-driven models have been developed to predict the evolution of LGG under chemotherapy [100]. Biomathematical modeling might also be helpful to simulate and compare the activity of different chemo-radiotherapy strategies in silico [101]. Finally, beyond the fact that LGG may acquire chemoresistance [102], Temozolomide can also induce hypermutation in a subset of tumors [103]. Identification of factors able to predict such mutational changes would be of utmost importance for selecting the appropriate treatment at the optimal time for each patient.

## 3. Predicting Neural Interindividual Variability and Its Changes over Time

Although linked, factors related to the clinical–psychological status of the patient and those related to the dynamics of neural network reconfiguration are here considered separately.

### 3.1. Factors Related to the Clinical–Psychological Status of the Patient

#### 3.1.1. Epilepsy

Seizure is the first symptom in LGG, leading to diagnosis in the vast majority of patients [104]. An epileptic symptomatology is linked to a better oncological outcome [10]. Interestingly, computational models have evidenced that the onset of seizures corresponds with a time point which may already represent the overcompensation stage of cerebral plasticity, depending on the tumor growth rate and its pattern of progression, in particular in the event of massive invasion of the white matter tracts [105]. Translation of these results into the clinical situation is another argument in favor of early surgery in asymptomatic patients, before the occurrence of epilepsy [74]. Indeed, seizures can have a negative impact on daily life, in particular by preventing driving (and indirectly employment) for medico-legal reasons [91,106]. Furthermore, epilepsy is mainly elicited by the diffusion of the tumoral cells at the periphery of the LGG (and not by the glioma core itself) [107]. This explains why larger surgical resection has a higher impact on epilepsy. Indeed, postoperative seizure control is more likely when EOR is ≥91% and/or residual tumor volume is ≤19 cc [108]. This also supports the suggestion of supratotal resection (with removal of the PTZ) for functional reasons (in addition to the improvement of oncological outcomes), i.e., with an optimization of QoL as a result of freedom from epilepsy [49].

Notably, important inter-individual variability has been observed, with about 15% of LGG patients experiencing intractable seizures, notably in temporal and/or paralimbic gliomas. In this situation, it has been proposed to remove the hippocampus, even if not invaded by the tumor according to preoperative MRI, since this can result in significant improvement of epilepsy control [109]. During follow-up, the reappearance of seizures may be correlated with LGG relapse, and may prompt clinicians to propose reoperation prior to MT [25]. When further resection is not possible due to diffusion within critical structures, for example within the Rolandic area which is very epileptogenic [110], adjuvant medical treatment can have an impact on seizures [111].

#### 3.1.2. Cognitive and Emotional Status

Because LGG patients are generally young and enjoy active lives at diagnosis, presenting no or only slight deficit at so-called “standard neurological examination”, it has been claimed in the literature that these patients do not exhibit any significant functional disturbances [6]. However, important variations are found across patients in terms of their neurocognitive status. In fact, a recent cohort including 157 LGG patients who benefited from extensive neuropsychological evaluation before treatment showed that 55.4% of them had already experienced cognitive decline, in particular with respect to language, verbal episodic memory, psychomotor speed, attention, and executive functions (phonological and categorical fluency) [4]. Interestingly, neurocognitive impairments have also been found in patients with incidental LGG [112].

Such disturbances are mostly due to tumor dissemination along the white matter pathways. Indeed, domain-specific deteriorations have been linked to the involvement of neural networks subserving the corresponding cognitive functions, including, for example, verbal semantic decline correlated to the infiltration of the left inferior fronto-occipital fasciculus [113] or visuo-spatial impairments correlated to the invasion of the right superior longitudinal fasciculus [114]. As well as cognition, emotional status can also be altered by LGG, e.g., deficit of empathy due the invasion of the right arcuate fasciculus or cingulum [115,116], or heightened schizotypal traits related to damage of the left uncinate fasciculus [117]. This can result in changes of mood or even personality in LGG patients [118], which may explain possible changes in their social interactions [119] or sexual activity [120,121]. Therefore, in practice, movement and conation (i.e., the willingness which leads to action) [122], visuo-spatial cognition, language, executive functions, attention, memory, semantic processing [123,124], mentalizing, metacognition (knowing of knowing) [125], emotion, personality, and behavior [118] must be systematically and carefully evaluated in clinical routine at diagnosis, as well as before and after each treatment in LGG patients [4].

Remarkably, a correlation has recently been demonstrated between SNP variations in brain-derived neurotrophic factor, dopamine receptor 2, and catechol-O-methyltransferase, and neurocognitive function and ability to return to work (RTW) in glioma patients at diagnosis and at 3 months; patients with higher-performing alleles had better scores for neuropsychological testing [126]. These findings show that further investigations are needed to better understand the genetic contribution to neurocognition in LGG patients and their ability for functional compensation or recovery.

#### 3.1.3. Neurological Status

Due to earlier diagnosis of LGG, moderate or severe neurological deficits at clinical examination (e.g., hemiparesis or aphasia) are usually rare [17]. Beyond possible episodes of transient worsening which might be elicited by repeat seizures, permanent impairments that arise are generally due to voluminous gliomas with mass effect and/or to MT, with acceleration of the neoplasm kinetics. In other words, with more “prophylactic management”, such major deteriorations should cease to occur before the last stage of the disease, thus giving the opportunity for LGG patients to enjoy active lives for many years or even decades [7].

#### 3.1.4. Patients’ Needs

Definition of QoL is eminently variable from one human being to another. Indeed, beyond the fact that patients do not want to experience hemiplegia or aphasia, especially following surgical resection, their expectations are very different according to their lifestyles: do they work? (if yes, what kind of employment they have, and do they want to resume their professional activities postoperatively and/or during medical treatment?); what are their hobbies? (e.g., do they practice sports, art, etc.?); what is their socio-cultural environment? (e.g., do they speak multiple languages?) [127], do they need to drive? (if yes, what about possible medico-legal issues in case of visual field deficit and/or seizures?) [106]. On the basis of these individual wishes, with the aim of enabling each patient to develop long-term projects (such as getting married, having a baby, buying a house, etc.), and also with due consideration given to the neurocognitive assessment at diagnosis (i.e., the presence or otherwise of some degrees of disturbance), it is possible to elaborate tailored management “à la carte”, beginning with a selection of optimal tasks to be performed during awake surgery [128]. Moreover, therapeutic strategies should be re-adapted over time, not only according to the LGG course, but also taking into account possible changes in the patient’s priorities. An example of such an intra-individual variability could be when a patient would like to preserve executive functions during the first surgery because he or she was employed at the time, but sparing higher-order cognitive capacities is no longer absolutely mandatory a few years later because the patient has retired in the meantime. Therefore, a clear and extensive explanation of the principles of chronic tumoral disease should be provided to the patient and his or her relatives after the diagnosis. They should understand that management necessitates surveillance with constant anticipation of a specific lifelong therapeutic strategy, to give them the opportunity to make choices for their current and future life by thinking one step ahead—which increases the chance of finding a better psychological equilibrium [7,23].

### 3.2. Factors Related to the Dynamics of the Neural Network Reconfiguration

#### 3.2.1. Patterns of Neuroplasticity

The structural anatomy of the brain is highly variable across healthy individuals, especially at the cortical level [129], while variations are less pronounced at the level of the white matter tracts [18]. Furthermore, advances in non-invasive functional imaging methods which permit the investigation of the functional connectivity have resulted in the development of a large database demonstrating between-subject variability in the distribution of neural networks [130,131]. Recent models of neurocognition employed in basic neuroscience have rejected the classical localizationist dogma (one cerebral site underpinning one specific function), and go beyond a simple network organization of the central nervous system (one cerebral circuit underpinning one specific function) by evidencing the critical role of dynamic interplay within and across neural networks which allows behaviors to be constantly adapted to the surrounding world [132]. According to this meta-networking framework (based on a network of networks), complex cognitive abilities are made possible by the activation and coordination (combination or competition) of large-scale neural circuits involving domain-specific networks (e.g., movement or language circuits), and the activity of a multiple-demand system recruited during the performance of a wide range of cognitive-demanding activities with the aim of maintaining fluid intelligence [132,133].

Alongside this flexible and ever-changing physiological organization of the functional connectome, inter-individual anatomo-functional variability is significantly increased in brain-damaged patients, particularly in the event of slowly evolving lesions such as LGG [134]. These mechanisms of neuroplastic functional reshaping permit neurological compensation during LGG growth (explaining why the vast majority of patients are active at diagnosis), at least to some extent, considering that over half of LGG patients already experience some degree of cognitive disturbance at the first neuropsychological evaluation, as previously mentioned [4]. Maps of neuroplasticity have evidenced that potential for cortical reallocation is high (except for input such as the primary visual cortex and output such as the primary motor cortex), whereas axonal connectivity represents the main limit of functional reshaping [135,136,137]. Thus, as in healthy subjects, variability across LGG patients is greater for cortical than subcortical reorganization [17]. This implies that various dynamic processes of neural reconfiguration may be mobilized from one patient to another, such as peritumoral rearrangement, or recruitment of remote structures in the ipsilesional hemisphere and/or the contralateral side [21,22]. Inter-patient differences in such patterns of redistribution are strongly correlated with LGG characteristics, i.e., the volume of the tumor, the kinetics of the glioma (as plasticity is linked to the time course of the disease, with less compensation in more rapidly evolving lesions [134]), and to the severity of brain invasion, with less plastic potential in more diffuse tumors which migrate more widely along the white matter pathways [44,138]. These connectomal considerations play a pivotal role when selecting the optimal therapeutic attitude, knowing that better prediction of the individual processes of neural reconfiguration may be valuable for anticipating the next treatment(s), thereby avoiding missing the opportunity for action from an oncological point of view.

#### 3.2.2. Potential of Metaplasticity

Interestingly, it is also possible to observe variations in the pattern of cerebral reshaping in reaction to tumor relapse in the same patient over time. For example, despite a predominance of local recruitment around the LGG at diagnosis, compensatory mechanisms based on the involvement of more distant structures may be found at recurrence [22]. Such metaplasticity phenomena have been evidenced by longitudinal functional neuroimaging studies (performed before and after surgery) [23] as well as by changes in the intraoperative electrical mapping that occurred between the first resection and the second or even third operation(s) [12,138].

As a step forward, it has been suggested to reorient the processes underlying metaplasticity, for instance to induce a switch from a perilesional to a contra-hemispheric compensatory pattern with mobilization of the homotopic areas [139]. This could increase the EOR while preserving QoL during reintervention(s) [25]. To achieve this goal, personalized programs of functional rehabilitation might be elaborated based on an improved comprehension of the meta-network, especially by reinforcing the intercommunication across neural circuits—for example between the executive control network and the language network—to compensate for LGG and its resection within the traditionally so-called “language areas” (such as Broca’s area) [139]. Transcranial stimulation could potentiate such changes in individual plastic modalities [140].

#### 3.2.3. Therapeutic Factors and the Connectome

A better knowledge of neural processing offers new methods for the adaptation of therapeutic approaches in LGG patients.

Surgery: The classical glioma-based resection aiming at removing a tumor-mass invaginated in the brain as if well-delineated [141] (which by definition is not the case in diffuse LGG), must definitely be replaced by a connectome-based surgical approach under the guidance of cortico-subcortical electrical mapping in awake patients [142]. The main goal is to remove a maximal part of the brain invaded by invasive tumoral disease (including the PTZ in order to achieve a supratotal excision if functionally applicable), taking into consideration not only the glioma characteristics but also the individual pattern of functional rearrangement that occurred before the operation [24]. To optimize surgical selection and planning, such plastic processes can be investigated by combining the results of neurologic and neuropsychological assessments with those of the functional imaging obtained before surgery [21]. The needs of the patient will determine the tasks administrated during the awake mapping and cognitive monitoring in the operating theater [128]. In addition, in order to mimic real life, particular intraoperative protocols have been proposed, especially based on constant multitasking with time constraint resulting in an increase of cognitive demand during the resection [143]. Postoperative functional rehabilitation (program, timing, duration, etc.) should also be pre-planned before the operation, in order for the patient and his or her family to organize appropriately in advance (in particular, arranging to stop work for a few weeks). This patient-specific surgical approach has led to minimization of severe permanent deficits (less than 1% occurrence), preservation or even improvement of postoperative cognitive scores, with RTW in over 94% of LGG patients [4,90,91,144]. Notably, RTW is a critical endpoint which has been severely neglected in the literature, and must be evaluated more systematically in future studies [145]. The same reasoning can be applied for reoperation(s), when it should be envisioned that all the individual parameters may have changed in the meantime [25]. For example, if the LGG relapse is more bulky and cortically-located in a patient with no or only mild functional deterioration, the chances of achieving an improved resection are higher, prompting repeat surgery to be proposed [138]—while preserving higher-order functions [146].To sum up, the oncological purpose of LGG surgery aiming to achieve greater EOR does not negatively impact neurocognitive outcomes [90], and can even improve QoL because it is correlated to a better control of epilepsy in LGG [108]. Indeed, even in the event of gliomas involving brain regions which had long been deemed inoperable, such as the insular lobe, if one cannot complete a total resection during a first surgery to avoid functional complications, reoperation should be proposed before transformation to a secondary high-grade glioma. Repeat surgery can lead to optimization of the EOR, with maintenance of the neurological capacities owing to neuroplastic processes that have occurred in the meantime. Interestingly, based on the concept of connectome-guided surgery, resection probability maps have been built, which allow accurate preoperative estimation of EOR and residual tumoral volume, including by junior neurosurgeons [147,148,149]. Furthermore, by using network-level approaches, machine learning has recently demonstrated the capability to predict individual cognitive outcomes following LGG removal [150]. These new tools may enable a more reliable estimation of the onco-functional balance of (re)operation for each patient, thus refining surgical decision.Chemotherapy: Beyond the oncological considerations previously discussed, from a functional perspective, chemotherapy is particularly appropriate for very diffuse LGG with a predominant migratory pattern within the subcortical connectivity, i.e., with a lower chance of radical resection in case of (re)operation [25,138]. This is particularly true if the patient already exhibits significant neurocognitive (and a fortiori neurologic) disorders, demonstrating that mechanisms of brain reorganization have been overwhelmed [44]. Indeed, chemotherapy may be administrated with preservation of cognitive abilities and QoL [151].Radiotherapy: As evidenced by many studies involving objective neuropsychological evaluation performed within 18 months [152,153] and after long-term follow-up following brain RT [94,154], there is a high risk of inducing permanent cognitive decline with a negative impact on QoL, including possible dementia, even if the glioma is still under control [94]. Importantly, as well as possibly inhibiting hippocampal neurogenesis, radiation-induced impairment of cognitive abilities is mainly related to white matter tract damage [155,156,157]. Therefore, beyond the classical optic pathways and the hippocampus, further organs at risk should be defined at the individual level, on the basis of an improved understanding of the functional connectome, its potential, and the limits of reconfiguration [158]; it is puzzling to note that this critical factor is not considered in the current guidelines [62].Combination of treatments: Because the aim is to predict how the glioma will progress and how the individual therapeutic strategy should be designed accordingly, various combinations of treatments can be proposed by taking into account not only oncological but also functional aspects. To this end, postponement of adjuvant medical therapies after surgery should be discussed more systematically, notably following radical resection, particularly to defer RT due to the neurocognitive risks in patients with long life expectancy. For example, in the event of foci of MT within the LGG, from a purely oncological perspective it might seem reasonable to perform chemo-radiotherapy immediately after surgery [83]. However, from a more integrated point of view aiming at optimizing the onco-functional balance (i.e., to influence positively the glioma course while preserving long-term QoL [159]), regular surveillance after extensive connectome-based resection may represent a more appropriate attitude—as confirmed by a 5-year survival rate of >95% with most surviving patients still active professionally, without seizures [87]. The same reasoning can be made in other “high-risk” LGG cases, e.g., to postpone RT in patients older than 40 years, on the condition that at least a subtotal resection was achieved and that pre- and post-operative LGG kinetics are slow [7], or even after maximal resection in glioma with an unfavorable molecular pattern [63]. Conversely, when reoperation(s) cannot be performed because of the invasion of the deep connectivity, as mentioned, upfront chemotherapy may be envisioned [97,98].

In summary, it is nowadays difficult to refer to “personalized medicine” as management based almost entirely on the molecular pattern, as proposed in the current recommendations [62], with little consideration paid to the dynamics of the functional brain for each patient at each stage of the disease.

## 4. Discussion

Beyond a better understanding of each factor taken in isolation, a comprehensive capture of the interplay between changes in LGG behavior and changes in neural network organization, acquired at the individual level over several years, is necessary to tailor a personalized longitudinal therapeutic strategy. To this end, real-time adaptation to the disease progression is insufficient since the diagnosis of LGG is typically made long after the occurrence of the tumor. The new concept requires thinking one step (or several steps) ahead, to (re)treat “preventively” over years or decades, with the aim of avoiding excessive diffusion of the glioma within the connectome, to prevent MT and to increase OS while preserving the functional status of the patient despite regular treatments. Importantly, considering the constantly changing mosaic of individual predictive factors (beyond the traditional crude markers such age < or ≥40 years, tumor size based on only one dimension, presence of neurologic deficits at standard clinical examination, and astrocytic histology [28,29,30]) it may be valuable to forge in advance a holistic view of the cross-talk between LGG and the individual brain and its changes. This can keep open the possibility of using the optimal treatment(s) in future, because the therapeutic armamentarium for LGG remains limited and some therapies cannot be repeated, such as reoperation within the subcortical connectivity or RT in voluminous gliomas. In other words, different management plans should be considered after diagnosis, and adjusted along the way, rather than applying the current guidelines that (i) are based only on a few parameters (mostly molecular profile for LGG); (ii) include few considerations regarding long-term outcomes (especially with respect to cognitive status, usually not evaluated after 5 years following therapy in patients who survive more than 15 years); and (iii) are due to undergo significant changes in a few years (with the constant modifications of the WHO classification).

## 5. Conclusions

In summary, surgical and medical oncologists have for a long time managed LGG patients from a step behind, due to late discovery of this poorly known tumoral disease, and major interindividual variability at a given moment and throughout progression of the disease. It is urgent that clinicians find ways to catch up and gain an advantage over the glioma. As well as intrinsic tumoral factors such as glioma volume, kinetics, metabolism, and genetics, interactions with non-oncological parameters such as SNPs, degree of brain reorganization and its impact on the patient’s behavior, or occurrence of pregnancy (to name only a few) should be accurately investigated in a systematic manner over several years, in order to anticipate tailored multistep management based on the predicted influence of therapies (in isolation and/or in association) on the neoplasm and the neural networks. Due to the multiplicity of these criteria and the complexity of their ever-changing intercommunications, deep-learning models could play a role in comparing in silico the efficacy of various treatment approaches, aiming to determine the strategy that will enable not only the optimization of long-term OS but also the preservation of QoL in agreement with the expectations of each human being experiencing LGG.

## Figures and Tables

**Figure 1 jpm-12-01621-f001:**
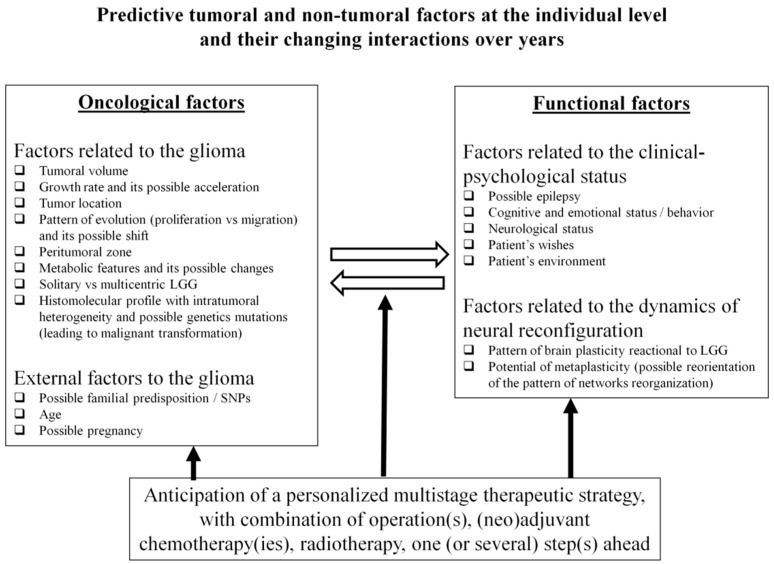
Predictive tumoral and non-tumoral factors at the individual level and their changing interactions over years.

## Data Availability

Not applicable.

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
