# Peer review of "A Personalized Longitudinal Strategy in Low-Grade Glioma Patients: Predicting Oncological and Neural Interindividual Variability and Its Changes over Years to Think One Step Ahead"

_jpm, 2022, doi:10.3390/jpm12101621_

Round 1

Reviewer 1 Report

This manuscript gives comprehensive review of the factors concerning the personalized LGG patient care with longitudinal strategy. The manuscript is well written and interesting to read. I do not see significant errors that need to be corrected.

Author Response

I thank the Reviewer for his/her positive comments.

Reviewer 2 Report

This is a nice summary of LGG from different aspects including diagnosis, factors related glioma progression and treatments. When and how to resect LGG is always a difficult decision for surgeons. As we know most surgeons now have given up "wait and see" opinion on LGG. But those tumors involved insular lobe, brain stem; if one can not make a total resection for the first surgery, the tumor may relapse and induce a secondary high grade glioma. But if you remove it radically, patients may have server complications. Please make comments or discuss this issue.

Author Response

I thank the Reviewer 2 for his/her constructive comment.

In the new version of the manuscript, in the paragraph 3.1.3. Therapeutic factors and the connectome (page 12), it has been added that :

« To sum up, the oncological purpose of LGG surgery aiming to achieve a greater EOR does not negatively impact neurocognitive outcomes [90], or even can improve QoL because correlated to a better control of epilepsy in LGG [108]. Indeed, even in the event of gliomas involving brain regions which were deemed inoperable for a long time, such as the insular lobe, if one can not make a total resection during a first surgery to avoid functional complications, reoperation should be proposed before transformation to a secondary high-grade glioma: repeat surgery can lead to an optimization of the EOR with maintenance of the neurological capacities owing to neuroplastic processes that occurred in the meantime. »